# Deep Curvature Suite

## Abstract

The curvature of the loss, provides rich information on the geometry underlying neural networks, with applications in second order optimisation and Bayesian deep learning. However, accessing curvature information is still a daunting engineering challenge, inaccessible to most practitioners. We hence provide a software package the **Deep Curvature Suite**, which allows easy curvature evaluation for large modern neural networks. Beyond the calculation of a highly accurate moment matched approximation of the Hessian spectrum using Lanczos, our package provides: extensive *loss surface visualisation*, the calculation of the *Hessian variance* and *stochastic second order optimisers*. We further address and disprove many common misconceptions in the literature about the Lanczos algorithm, namely that it learns eigenvalues from the top down. We prove using high dimensional concentration inequalities that for specific matrices a single random vector is sufficient for accurate spectral estimation, informing our spectral visualisation method. We showcase our package practical utility on a series of examples based on realistic modern neural networks such as the VGG-16 and Preactivated ResNets on the CIFAR-10/100 datasets. We further detail 3 specific potential use cases enabled by our software: research in stochastic second order optimisation for deep learning, learning rate scheduling using known optimality formulae for convex surfaces and empirical verification of deep learning theory based on comparing empirical and theoretically implied spectra.

## 1 Introduction

The success of deep neural networks trained with gradient based optimisers in speech and object recognition (LeCun et al., 2015), has led to an explosion in easy to use high performance software implementations. Automatic differentiation packages such as TensorFlow (Abadi et al., 2016) and PyTorch (Paszke et al., 2017) have become widely adopted. Higher level packages, such as Keras (Chollet, 2015) allow practitioners users to state their model, dataset and optimiser in a few lines of code, effortlessly achieving state of the art performance.

However, software for extracting second order information, representing the curvature of the loss at a point in weight space, has not kept abreast. Researchers aspiring to evaluate curvature information need to implement their own libraries, which are rarely shared or kept up to date. Naive implementations, which rely on full eigendecomposition (cubic cost in the parameter count) are computationally intractable for all but the smallest of models. Hence, researchers typically ignore curvature information or use highly optimistic approximations. Examples in the literature include the diagonal elements of the matrix or of a surrogate matrix,Chaudhari et al. (2016); Dangel et al. (2019), which we show in AppendixE can be very misleading.

## 2 Motivation

The curvature of the loss informs us about the local conditioning of the problem (i.e the ratio of the largest to smallest eigenvalues $\frac{\lambda_1}{\lambda_P}$). This determines the rate of convergence for first order methods and informs us about the optimal learning and momentum rates (Nesterov, 2013). Hence easily accessible curvature information could allow practitioners to scale their learning rates in an optimal way throughout training, instead of relying on expert scheduling, we investigate this using our software in Section 5.2. Research areas where curvature information features most prominently are analyses of the **Loss Surface** and **Newton** type optimization methods.

## 2.1 Loss Surfaces

Recent neural network loss surfaces using full eigendecomposition (Sagun et al., 2016; 2017) have been limited to toy examples with less than five thousand parameters. Hence, loss surface visualisation of deep neural networks have often focused on two dimensional slices of random vectors (Li et al., 2017) or the changes in the loss traversing a set of random vectors drawn from the $d$-dimensional Gaussian distribution (Izmailov et al., 2018). It is not clear that the loss surfaces of modern expressive neural networks, containing millions or billions of dimensions, can be well captured in this manner. Small experiments have shown neural networks have a large rank degeneracy Sagun et al. (2016) with a small number of large outliers. However high dimensional concentration theorems Vershynin (2018) guarantee that even a large number of randomly sampled vectors are unlikely to encounter such outliers and hence have limited ability to discern the geometry between various solutions. Other works, that try to distinguish between flat and sharp minima have used the diagonal of the Fisher information matrix (Chaudhari et al., 2016), an assumption we will challenge in this paper. Specifically we show in Appendix E, that diagonal approximations do not capture key properties in synthetic and real neural network examples.

From a practical perspective, specific properties of the loss surface are not captured by the aforementioned approaches. Examples include the *flatness* as specified by the trace, Frobenius and spectral norm. These measures have been extensively used to characterise the generalisation of a solution found by SGD (Wu et al., 2018; Izmailov et al., 2018; He et al., 2019; Jastrzebski et al., 2017; 2018; Keskar et al., 2016). Under a Bayesian and minimum description length argument (Hochreiter and Schmidhuber, 1997) flatter minima should generalise better than sharper minima. The magnitude of these outliers have been linked to poor generalisation performance Keskar et al. (2016) and as a consequences the generalisation benefits of large learning rate SGD Wu et al. (2018); Jastrzebski et al. (2017). These properties are extremely easy in principle to estimate, at a computational cost of a small multiple of gradient evaluations. However the calculation of these properties are not typically included in standard deep learning frameworks, which limits the ability of researchers to undertake such analysis.

Other important areas of loss surface investigation include understanding the effectiveness of batch normalisation(Ioffe and Szegedy, 2015). Recent convergence proofs (Santurkar et al., 2018) bound the maximal eigenvalue of the Hessian with respect to the activations and bounds with respect to the weights on a per layer basis. Bounds on a per layer basis do not imply anything about the bounds of the entire Hessian and furthermore it has been argued that the full spectrum must be calculated to give insights on the alteration of the landscape (Kohler et al., 2018).

## 2.2 Second Order Optimisation Methods

Second order optimisation methods solve the minimisation problem for the loss, $L(\boldsymbol{w}) \in \mathbb{R}$ associated with parameters $\boldsymbol{w} \in \mathbb{R}^{P \times 1}$ and perturbation $\boldsymbol{\delta w} \in \mathbb{R}^{P \times 1}$ to the second order in Taylor expansion,

$$\boldsymbol{\delta w}^* = \text{argmin}_{\boldsymbol{\delta w}} L(\boldsymbol{w} + \boldsymbol{\delta w}) = \text{argmin}_{\boldsymbol{\delta w}} L(\boldsymbol{w} + \boldsymbol{\delta w}) = -\bar{\boldsymbol{H}}^{-1} \nabla L(\boldsymbol{w}). \tag{1}$$

Sometimes, such as in deep neural networks when the Hessian $\boldsymbol{H} = \nabla^2 L(\boldsymbol{w}) \in \mathbb{R}^{P \times P}$ is not positive definite, a positive definite surrogate is used. Note that Equation 1 is not lower bounded unless $\boldsymbol{H}$ is positive semi-definite. Often either a multiple of the identity is added (known as damping) to the Hessian ($\boldsymbol{H} \to \boldsymbol{H} + \gamma \boldsymbol{I}$) Dauphin et al. (2014) or a surrogate positive definite approximation to the Hessian $\bar{\boldsymbol{H}}$, such as the Generalised Gauss-Newton (GGN) (Martens, 2010; Martens and Sutskever, 2012) is employed. To derive the GGN, we express the loss in terms of the activation $L(\boldsymbol{w}) = \sigma(f(\boldsymbol{w}))$ of the output of the final layer $f(\boldsymbol{w})$. Hence the elements of the Hessian can be written as

$$\boldsymbol{H}(\boldsymbol{w})_{ij} = \sum_{k=0}^{d_y} \sum_{l=0}^{d_y} \frac{\partial^2 \sigma(f(\boldsymbol{w}))}{\partial f_l(\boldsymbol{w}) \partial f_k(\boldsymbol{w})} \frac{\partial f_l(\boldsymbol{w})}{\partial w_j} \frac{\partial f_k(\boldsymbol{w})}{\partial w_i} + \sum_{k=0}^{d_y} \frac{\partial \sigma(f(\boldsymbol{w}))}{\partial w_j} \frac{\partial^2 f_k(\boldsymbol{w})}{\partial w_j \partial w_i}. \tag{2}$$

The first term on the LHS of Equation 2 is known as the *Generalised Gauss-Newton* matrix.

Despite the success of second order optimisation methods using the GGN for difficult problems on which SGD is known to stall, such as recurrent neural networks (Martens and Sutskever, 2012), or

auto-encoders (Martens, 2016). Researchers wanting to implement second order methods such as (Vinyals and Povey, 2012; Martens and Sutskever, 2012; Dauphin et al., 2014) face the aforementioned problems of difficult implementation. As a small side note, Bayesian neural networks using the Laplace approximation feature the Hessian inverse multiplied by a vector (Bishop, 2006).

## 2.3 CONTRIBUTIONS

We introduce our package, the **Deep Curvature suite**, a software package that allows analysis and visualisation of deep neural network curvature. The main features and functionalities of our package are:

- **Eigenspectrum analysis of the curvature matrices** Powered by the Lanczos Meurant and Strakoš (2006) techniques implemented in GPyTorch (Gardner et al., 2018) and outlined in Section 3, *with a single random vector* we use the Pearlmutter matrix-vector product trick Pearlmutter (1994) for fast inference of the eigenvalues and eigenvectors of the common curvature matrices of the deep neural networks. In addition to the standard Hessian matrix, we also include the feature for inference of the eigen-information of the Generalised Gauss-Newton matrix, a commonly used positive-definite surrogate to Hessian[1].

- **Advanced Statistics of Networks** In addition to the commonly used statistics to evaluate network training and performance such as the training and testing losses and accuracy, we support computations of more advanced statistics: For example, we support squared mean and variance of gradients and Hessians (and GGN), squared norms of Hessian and GGN, L2 and L-inf norms of the network weights and etc. These statistics are useful and relevant for a wide range of purposes such as the designs of second-order optimisers and network architecture.

- **Visualisations** For all main features above, we include accompanying visualisation tools. In addition, with the eigen-information obtained, we also feature visualisations of the **loss landscape** by studying the sensitivity of the neural network to perturbations of weights.

- **Second Order Optimisation** We include out of the box stochastic lanczos second order optimisers which use the absolute Hessian Dauphin et al. (2014) or the Generalised Gauss Newton Martens (2010) as curvature matrices

- **Batch Normalisation Support** Our code allows for the use of batch normalisation(Ioffe and Szegedy, 2015) in neural networks, which is integrated by default on most modern convolutional neural networks. Since the way in watch batch normalisation is calculated has a big effect on the resulting curvature calculations and can be dependent on the order in which the samples are processed, we offer two models of evaluation. *Train mode* which depends on the order in which the samples are processed and is similar to what the optimiser sees during training and *evaluation mode*, in which the statistics are pre-computed and hence invariant to sample shuffling. We expand on the differences between these methods in Appendix A.

Our software makes calculating and visualising curvature information as simple as calculating the gradient at a saved checkpoint in weight-space. The computational complexity of our approach is $\mathcal{O}(mP)$, where $m$ is the number of Lanczos steps [2] and $P$ is the number of model parameters. This in stark contrast full exact eigendecomposition which has a numerical cost of $\mathcal{O}(P^3)$, which is infeasible for large neural networks.

## 2.4 RELATED WORK

Recent work making curvature information more available using diagonal approximations, disallows the use of batch normalisation(Dangel et al., 2019). Our software extends seamlessly to support

---

[1]The computation of the GGN-vector product is similar with the computational cost of two backward passes in the network. Also, GGN uses *forward-mode automatic differentiation* (FMAD) in addition to the commonly employed *backward-mode automatic differentiation* (RMAD). In the current PyTorch framework, the FMAD operation can be achieved using two equivalent RMAD operations.

[2]which corresponds to the number of moments of the underlying Hessian/GGN density which we wish to match

| Functionality | Deep Curvature Suite | PyHessian |
|---|---|---|
| Hessian Density Estimation | ✓ | ✓ |
| GGN Density Estimation | ✓ | ✗ |
| Loss Surface Visualisation | ✓ | ✓ |
| Gradient Variance | ✓ | ✗ |
| Hessian/GGN Variance | ✓ | ✗ |
| Second Order Optimisers | ✓ | ✗ |

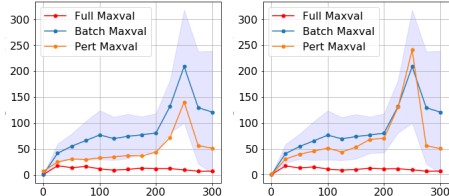

Table 1: Table of Contributions: comparison between our released software and that of another PyTorch based Hessian tool PyHessian

Figure 1: Deviation of the mini-batch Hessian (left)/GGN (right) spectral norm from the full dataset, Figure from Granziol (2020).

batch normalization In Appendix E.5 we explicitly compare against their diagonal MC approximations on synthetic and real neural network examples and find these approximations inadequate for spectral analysis. Independent work has also uses the Lanczos algorithm for Hessian computation Yao et al. (2019); Ghorbani et al. (2019); Papyan (2018); Izmailov et al. (2019). We compare the functionality of our package to that of PyHessian Yao et al. (2018) in Table 1.

One major extra functionality we offer is the calculation of the GGN. For common activatios and loss functions, such as the cross-entropy loss and sigmoid activation, the generalised Gauss-Newton is equivalent to the Fisher information matrix (Pascanu and Bengio, 2013). Hence this approximation to the Hessian is interesting in its own right and many theoretical analyses, consider the generalised Gauss-Newton or Fisher information matrices as opposed to the Hessian Pennington and Worah (2018); Karakida et al. (2019); Papyan (2019). Hence accessing the GGN spectrum could be a major asset to researchers. Another extra functionality we offer is the calculation of the Hessian/GGN variance. This quantity has been used to derive optimal learning rates in Wu et al. (2018) and to predict the effect of minibatching on curvature using a random matrix theory model (Granziol, 2020). We include an example figure where the theoretical prediction is validated using our software in Figure 1.

## 3 Lanczos, Misconceptions and Spectral Density Approximations

The Lanczos Algorithm Meurant and Strakoš (2006), on which our package is based, (Algorithm 1 in the Appendix) is an iterative algorithm for learning a subset of the eigenvalues/eigenvectors of any Hermitian matrix, requiring only matrix vector products. It is an adaptation of the power iteration method, where the Krylov subspace $\mathscr{K}(\boldsymbol{H}, \boldsymbol{v}) = \text{span}\{\boldsymbol{v}, \boldsymbol{H}\boldsymbol{v}, \boldsymbol{H}^2\boldsymbol{v}, ...\}$, is orthogonalised using Gram-Schmidt, resulting in improved convergence. The Lanczos algorithm, is subject to many misconceptions in the literature, which limits its penetration in the research community. Two of which we state here and explicitly debunk in Appendix B.2

- We can learn the negative and interior eigenvalues by shifting and inverting the matrix sign $\boldsymbol{H} \to -\boldsymbol{H} + \mu\boldsymbol{I}$
- Lanczos learns the largest $m$ $[\lambda_i, \boldsymbol{u}_i]$ pairs of $\boldsymbol{H} \in \mathbb{R}^{P \times P}$ with high probability (Dauphin et al., 2014)

The key properties of the Lanczos algorithm, which we summarise here (and expand upon in Appendix B) are:

- When seeded with a vector $\boldsymbol{v}$ the Lanczos algorithm gives a discrete spectral density which matches the $m$ moments $\boldsymbol{v}\boldsymbol{H}\boldsymbol{v}, \boldsymbol{v}\boldsymbol{H}^2\boldsymbol{v}, ..., \boldsymbol{v}\boldsymbol{H}^m\boldsymbol{v}$.
- If the vector $\boldsymbol{v}$ is zero mean unit variance (such as a Gaussian or Rademacher Hutchinson (1990)), this in expectation gives an $m$ moment matched approximation to the spectral density $p(\lambda)$ of the underlying matrix $\boldsymbol{H}$.
- For a large dimensional matrix $P \to$, we expect a single random vector to give an $m$ moment matched approximation of the underlying spectral density. We give the proof for this in Appendix C.This informs our visualisation technique of using a single random vector.
- When seeded with a random vector $\boldsymbol{v}$, Lanczos converges quickly to the end points of the spectral support of $p(\lambda)$, the spectral density of $\boldsymbol{H}$.

## 4 AN ILLUSTRATED EXAMPLE

We give an illustration on an example of using the Deep Curvature package. We train a VGG16 network on CIFAR-100 for 100 epochs. With the checkpoints generated, we may now compute analyse the eigenspectrum of the curvature matrix evaluated at the desired point in training.To evaluate the Hessian/GGN with 100 Lanczos iterations, we run the code given in Table 2.

**Network Training**

```
train_network(
dir='VGG16/',
dataset='CIFAR100',
data_path='data/',
epochs=100, model='VGG16',
optimizer='SGD',
optimizer_kwargs=
{'lr': 0.03, 'mom': 0.9,
'weight_decay': 5e-4})
```

**Eigenspectrum Computation**

```
compute_eigenspectrum(
dataset='CIFAR100',
data_path='data/',
model='VGG16',
checkpoint_path='checkpoint.pt',
save_spectrum_path='spectrum-ggn',
save_eigvec=True,
lanczos_iters=100,
curvature_matrix='ggn_lanczos',)
```

Table 2: Training a Neural Network and Calculating the Eigenspectrum using the Deep Curvature Suite package

This function call saves the spectrum results (including eigenvalues, eigenvectors and other related statistics) in the save_spectrum_path path string defined. To visualise the spectrum as stem plot we simply run the code given in Table 3 with corresponding example Figure shown.

**Eigenspectrum Visualization**

```
plot_spectrum('lanczos',
path='spectrum-ggn.npz')
plt.show()
```

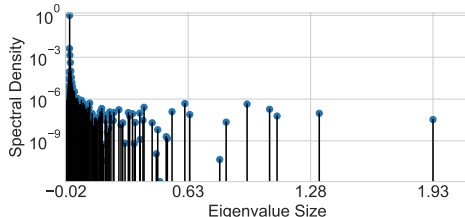

Table 3: Eigenspectrum plotting code and corresponding stem plot

Finally, with the eigenvalues and eigenvectors computed, we might be interested in knowing how sensitive the network is to perturbation along these directions. To achieve this, we first construct a loss landscape by setting the number of query points and maximum perturbation to apply. To achieve that, we call the code given in Table 4. In this example, we set the maximum perturbation to be 1

**Loss Surface**

```
build_loss_landscape(
dataset='CIFAR100',
data_path='data/',
model='VGG16',
dist=1., n_points=21,
spectrum_path=hessian',
checkpoint_path='ck.pt',
save_path='scape.npz'
plot_loss_landscape
('landscape-100.npz')
plt.show())
```

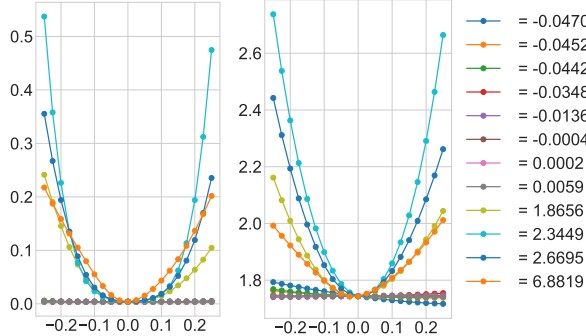

Table 4: Loss surface visualization along the sharpest Hessian eigenvectors with the Deep Curvature Suite

(dist argument) and number of query points along each direction to be 21 (n_points argument). The corresponding plots in Training and Testing loss are also shown. Although we choose to show loss, the package returns the results for accuracy also, which we show in Appendix D.

## 5 EXAMPLE RESEARCH APPLICATIONS

In this section we list 3 example scientific applications of our software package. We investigate second order optimisation for deep neural networks using our inbuild stochastic Lanczos optimiser, online learning rate scheduling and verifying predictions made using random matrix theory in theoretical deep learning.

### 5.1 SIMPLICITY OF SECOND ORDER OPTIMISATION WITH THE DEEP CURVATURE SUITE

To enable researchers to experiment with stochastic second order optimisation algorithms for deep neural networks, we implement a Lanczos based optimiser (which takes the absolute Hessian Dauphin et al. (2014), or Generalised Gauss Newton Martens (2010) as input). The code for running this optimiser is summarised in Table 5. We plot the training error of the VGG-16 network on

### SGD Training

```
train_network(
dir='VGG16–CIFAR100/',
dataset='CIFAR100',
data_path='data/',epochs=100,
model='VGG16', optimizer='SGD',
optimizer_kwargs={
'lr': 0.01,
'momentum': 0.9,
'weight_decay': 0
'batch_size': 128})
```

### Lanczos Training

```
train_network(
dir='VGG16–CIFAR100/',
dataset='CIFAR100',
data_path='data/',epochs=100,
model='VGG16', optimizer='LancGN',
optimizer_kwargs={
'lr': 1,
'damping': 10,
'weight_decay': 0
'batch_size': 128
'curvature_batch_size': 128})
```

Table 5: Comparison of SGD training and Second Order Optimisation using the Deep Curvature Suite

CIFAR-100 dataset, which has over 16 million parameters and hence is out of reach of full inversion methods, in Figure 2b. We keep the ratio of damping constant to learning rate constant, where $\delta = 10\alpha$, for a variety of learning rates in $\{1, 0.1, 0.01, 0.001, 0.0001\}$ with a batch size of 128 for both the gradient and the curvature, all of which post almost identical performance. We also compare against different learning rates of Adam and the best grid searched learning rate of SGD, both of which converge significantly slower per iteration compared to our stochastic Newton methods. We note that in terms of practical optimisation that the cost of running a stochastic second order optimiser is far greater than that of SGD. For this example, where we use 20 Lanczos iterations, the per iteration cost scales as $\approx 40$ times that of SGD. Hence we expect these algorithms simply to serve as useful baselines against more computationally efficient approximations. One example future direction could be to invert the matrices less frequently as done in Martens and Grosse (2015), reducing computational cost.

### 5.2 SPECTRAL STOCHASTIC GRADIENT DESCENT WITH THE DEEP CURVATURE SUITE

SGD is incredibly sensitive to the choice of initial learning rate and its scheduling. Whilst adaptive methods are considered more robust to the initial learning rate choice and its schedule, they have been shown to perform poorly on validation or held out test data Wilson et al. (2017). Hence practitioners typically experiment with a variety of schedules, such as the step, linear, exponential, polynomial and cosine annealing. One potential application of our codebase is to use a stochastic estimate of the curvature and to use established optimal learning and momentum rates for the local quadratic approximation at the given point in weightspace Nesterov (2013), given in Equation 3.

$$\alpha_{Polyak} = \frac{2}{\sqrt{\lambda_1} + \sqrt{\lambda_P}}, \rho = \left(\frac{\sqrt{\lambda_1} - \sqrt{\lambda_P}}{\sqrt{\lambda_1} + \sqrt{\lambda_P}}\right)^2, \alpha_{Nesterov} = \sqrt{\frac{\lambda_P}{\lambda_1}} \rho = \frac{\sqrt{\lambda_1} - \sqrt{\lambda_P}}{\sqrt{\lambda_1} + \sqrt{\lambda_P}}. \quad (3)$$

Where $\lambda_1, \lambda_P$ refer to the largest and smallest eigenvalues of the Hessian respectively. The learning rates and momenta $\alpha, \rho$ are given for both Polyak and Nesterov momentum respectively. There are several complications to the above method. Firstly the Hessian of neural networks is in general not

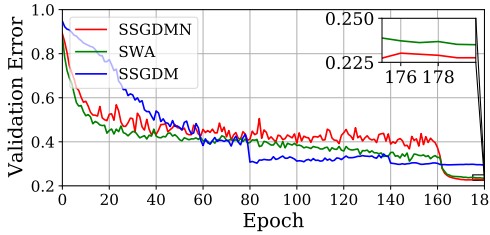 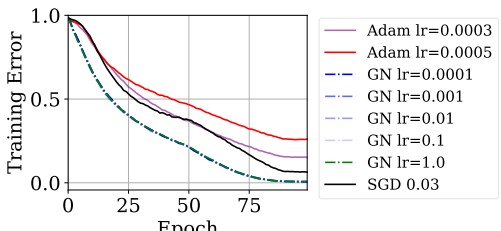

(a) Test Error CIIFAR-100 SSGDM(N) & SWA on the PreResNet-110          (b) Train Error VGG-16 CIFAR-100 Second Order Optimisers

positive definite and hence $\lambda_P < 0$ (these optimality formulae are strictly for convex methods). Secondly the loss surface at each point in weight space is likely to be different and it is impractical to recalculate the optimal step size using Equation 3 for every iteration, even if we sub-sample the data. We note from the work of Granziol (2020), that the loss surface when using a mini-batch varies to that of the full surface (it is sharper) and hence the learning rates when training using a mini-batch must be optimal for the minibatch surface and not that of the full dataset. As an example use case for our software we consider whether the method of Izmailov et al. (2018), which combines SGD with Polyak averaging Polyak and Juditsky (1992) at the end of training (which they denote *SWA*), to improve generalisation performance, could be improved. Instead of fine tuning the linear decay learning rate schedule by hand, we instead learn the learning rate using Equation 3 at regular intervals by estimating the curvature using a sub-sampled Lanczos spectral estimate.

We run the preactivated ResNet-110 on the CIFAR-100 dataset for 180 epochs and learn the Polyak and Nesterov learning rates and momenta every 20 epochs using the same mini-batch size as the gradient batch size 128 for 20 Lanczos iterations. Since the smallest Ritz values are very very close to 0, which would result in a momentum $\rho = 1$, we use a heuristic to remove the smallest Ritz values, whereby if the Ritz value of largest spectral mass has more than 50% of the spectral mass, it is removed and the resulting density renormalised, forming the new spectral density of interest. All methods employ Polyak Avearging at epoch 161 as in Izmailov et al. (2018). We plot the results in Figure 2a and show that the Nesterov variant (SSGDMN) of online learning rate and momentum learning performs well against the baseline (SWA). Interestingly the Polyak variant (SSGDM) learns to drop the learning rate at regular intervals, similar to classical training methods, which does not perform as well as keeping the learning rate high and using Polyak averaging at the end of training.

## 5.3   VERIFYING RANDOM MATRIX THEORY IN DEEP LEARNING USING OUR PACKAGE

The full spectrum of deep neural networks can be used to validate/invalidate novel theoretical contributions. As an example. analysis relating the loss surface to spin-glass models from condensed matter physics and random matrix theory (Choromanska et al., 2015b;a) rely on a number of unrealistic assumptions, such as input independence (i.e. for an input $x$ that the feature $x_i$ is independent of $x_j$, for images which have spatial correlations, such assumptions are clearly violated). One potential verification of the practical applicability of these results, that does not involve checking that the assumptions are satisfied, would be to visualise the spectra of large real networks and commonly used datasets. This follows because spin glass models have Hessians given by random matrices (Arous and Guionnet, 1997) with known analytical forms (Tao, 2012; Akemann et al., 2011). Hence statistical tests comparing the real world observed spectra and that underlying the specific spin glass model could be undertaken, showing that the results hold despite the assumptions not being met. Two primary random matrix eigenvalue distributions to which the spectra of many classes of random matrices converge (and which feature in theoretical deep learning papers) are the semi-circle and the Marchenko-Pastur densities defined below.

$$\overbrace{p(\lambda) = \frac{1}{2\pi}\sqrt{4\sigma^2 - \lambda^2}\mathbf{1}_{|\lambda| \leq 2\sigma}}^{\text{elements } \boldsymbol{H}_{i,j} \text{ are i.i.d normal}}, \quad \overbrace{p(\lambda) = \frac{\sqrt{(\lambda - \lambda_+)(\lambda - \lambda_-)}}{2\pi\lambda q\sigma^2}}^{\text{elements } \boldsymbol{X}_{i,j} \text{ are i.i.d normal } \boldsymbol{H} = \boldsymbol{X}^T\boldsymbol{X}}, \lambda_\pm = \sigma^2(1 \pm \sqrt{q})^2 \quad (4)$$

$$\underbrace{\phantom{p(\lambda) = \frac{1}{2\pi}\sqrt{4\sigma^2}}}_{\text{Semi-Circle Law}} \quad \underbrace{\phantom{p(\lambda) = \frac{\sqrt{(\lambda)}}{2\pi\lambda}}}_{\text{Marchenko-Pastur}}$$

If there are $N$ samples and $P$ parameters and $N < P$ the Marchenko-Pastur density has a degeneracy fraction $1 - \frac{N}{P}$.

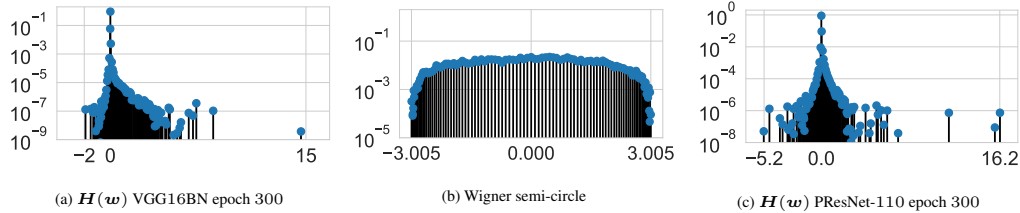

(a) $\boldsymbol{H}(\boldsymbol{w})$ VGG16BN epoch 300

(b) Wigner semi-circle

(c) $\boldsymbol{H}(\boldsymbol{w})$ PResNet-110 epoch 300

Figure 3: Neural network Hessian spectra at the end of training for the VGG-16BN and PreResNet-110 on the CIFAR-100 dataset and an example of the semi-circle law

In the example of Choromanska et al. (2015a), the corresponding Hessian spectral denisty is the semi-circle law, shown in Figure 3b. This differs greatly from neural network spectra found using our package, such as that of the VGG-16 with BN shown in Figure 3a and the Preactived ResNet-110, for the CIFAR-100 datasets, shown in Figure 3c. Major differences include the rank-degeneracies (we find that over $90\%$ of the spectral mass resides in the peak nearest the origin) and outliers. The proof techniques used in Arous and Guionnet (1997), which underlie the claims in Choromanska et al. (2015a) specifically use the large deviation principle of the Gaussian Orthogonal Ensemble (which converges to the semi-circle law) and hence forbid outliers.

The GGN has been modelled as the Marchenko-Pastur density in the literature (Pennington and Bahri, 2017; Sagun et al., 2017; 2016). The Residual matrix $\mathcal{R}(\boldsymbol{w})$, which is the difference of the Hessian and the GGN, has been modelled by the Wigner ensemble which gives the semi-circle law in Pennington and Bahri (2017). In Figure 4 we plot the GGN and Residual of a VGG-16 network on the CIFAR-100 dataset. The GGN unlike the Marchenko-Pastur has no spectral gap (the density tails off exponentially) and many outliers. Similarly the shape and rank-degeneracy of the Residual matrix varies significantly from the semi-circle and contains outliers. Whilst a single set of negative results does not invalidate an entire framework, it implies that further research is required to extend the claims of Choromanska et al. (2015a); Pennington and Bahri (2017) to practical deep learning. These results using our package demonstrate that the underlying spectral densities of the matrices in question vary considerably from those given by the theoretical predictions. This implies that the conclusions made may not necassarily hold in real neural networks and requires further investigation.

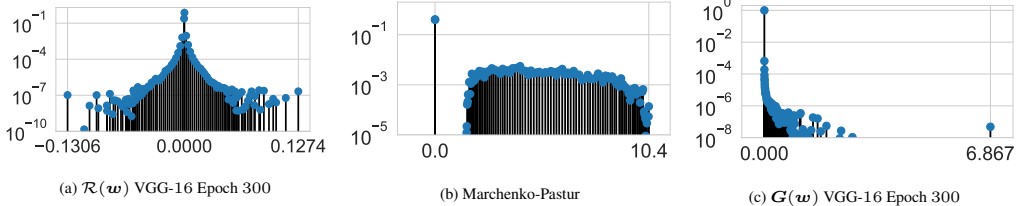

(a) $\mathcal{R}(\boldsymbol{w})$ VGG-16 Epoch 300

(b) Marchenko-Pastur

(c) $\boldsymbol{G}(\boldsymbol{w})$ VGG-16 Epoch 300

Figure 4: The Generalised Gauss Newton and Residual Matrix at various Epochs of Training for the VGG-16 on CIFAR-100 and a sample Marcenko-Pastur density

## 6 CONCLUSION

We introduce the **Deep Curvature** suite in **PyTorch** framework, based on the Lanczos algorithm implemented in GPyTorch (Gardner et al., 2018), that allows deep learning practitioners to learn spectral density information as well as eigenvalue/eigenvector pairs of the curvature matrices at specific points in weight space. Our package also allows the user to evaluate the Generalised Gauss Newton spectral density, the variance of both the Hessian and the Generalised Gauss Newton and includes two stochastic Lanczos based optimisers. Together with the software, we also include a succinct summary of the linear algebra, iterative method theory including proofs of convergence and misconceptions and stochastic trace estimation that form the theoretical underpinnings of our work. We also give 3 example scientific applications of our work, namely second order optimisation on large scale neural networks, learning rate scheduling using spectral information and evaluation of theoretical results which make predictions on spectral denisities of the Hessian of the neural networks which they are modelling.

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

## A    Subtleties of Batch Normalisation

Most modern convolutional neural networks use batch normalisation Ioffe and Szegedy (2015). Whilst we refer the reader to Ioffe and Szegedy (2015) for a full description of batch normalisation, a subtlety which is introduced is that the output of a network for one sample depends on the samples in the mini-batch (as they are used to compute the statistics). Since these statistics are used to normalise and center the layers, they have a large effect on the curvature calculation. In our software package, we give the user two options to deal with this. We denote one **Train Mode** and the other **Evaluation Mode**. We detail them both below

### A.1    Train Mode

With batch norm in train mode the output of the network for one sample depends on other samples in the mini-batch. This effect influences the result of hessian-vector product computation on a dataset. In particular, if the order of samples in the dataset is changed, then the set of mini-batches also changes and the hessian-vector product will be different. We take this effect into account and fix the order of samples in the dataset, so that hessian-vector product for different networks is computed on the same set of mini-batches. Conceptually, one can use the following interpretation of a neural network with batch norm layers in train mode: the input of such a network is not a single sample but a mini-batch of samples. Instead of thinking about a dataset of samples we should think about a dataset of mini-batches. With the fixed order of samples, we guarantee that all models are evaluated on the same dataset of mini-batches.

### A.2    EvaluationMode

One can also use BN layers in eval mode with BN statistics computed over the whole dataset. With BN layers in eval mode the output of a network on sample does not depend on other samples in the mini-batch. Our code the mode of BN layers can be specified as a parameter for hessian-vector product computation.

## B    Learning to love Lanczos

The Lanczos Algorithm, on which our package is based, (Algorithm 1) is an iterative algorithm for learning a subset of the eigenvalues/eigenvectors of any Hermitian matrix, requiring only matrix vector products. It can be regarded as a far superior adaptation of the power iteration method, where the Krylov subspace $\mathcal{K}(\boldsymbol{H}, \boldsymbol{v}) = \text{span}\{\boldsymbol{v}, \boldsymbol{H^v}, \boldsymbol{H}^2\boldsymbol{v}...\}$ is orthogonalised using Gram-Schmidt. Beyond having improved convergence to the power iteration method (Bai et al., 1996) by storing the intermediate orthogonal vectors in the corresponding Krylov subspace, Lanczos produces estimates of the eigenvectors and eigenvalues of smaller absolute magnitude, known as Ritz vectors/values. Despite its known superiority to the power iteration method and relationship to orthogonal polynomials and hence when combined with random vectors the ability to estimate the entire spectrum of a matrix, these properties are often ignored or forgotten by practitioners. We compare against diaognal approximations for random and synthetic matrices in Appendix E.

The Lanczos method be be explicitly derived by considering the optimization of the Rayleigh quotient (Golub and Van Loan, 2012)

$$r(\boldsymbol{v}) = \frac{\boldsymbol{v}^T \boldsymbol{H} \boldsymbol{v}}{\boldsymbol{v}^T \boldsymbol{v}} \tag{5}$$

over the entire Krylov subspace $\mathcal{K}_m(\boldsymbol{H}, \boldsymbol{v})$ as opposed to power iteration which is a particular vector in the Krylov subspace $\boldsymbol{u} = \boldsymbol{H}^m\boldsymbol{v}$. Despite this, practitioners looking to learn the leading

| $\lambda_1/\lambda_2$ | $m = 5$ | $m = 10$ | $m = 15$ | $m = 20$ |
|---|---|---|---|---|
| 1.5 | $\frac{1.1\times10^{-4}}{3.9\times10^{-2}}$ | $\frac{2\times10^{-10}}{6.8\times10^{-4}}$ | $\frac{3.9\times10^{-16}}{1.2\times10^{-5}}$ | $\frac{7.4\times10^{-22}}{2.0\times10^{-7}}$ |
| 1.1 | $\frac{2.7\times10^{-2}}{4.7\times10^{-1}}$ | $\frac{5.5\times10^{-5}}{1.8\times10^{-1}}$ | $\frac{1.1\times10^{-7}}{6.9\times10^{-2}}$ | $\frac{2.1\times10^{-10}}{2.7\times10^{-2}}$ |
| 1.01 | $\frac{5.6\times10^{-1}}{9.2\times10^{-1}}$ | $\frac{1.0\times10^{-1}}{8.4\times10^{-1}}$ | $\frac{1.5\times10^{-2}}{7.6\times10^{-1}}$ | $\frac{2.0\times10^{-3}}{6.9\times10^{-1}}$ |

Table 6: $L_{k-1}/R_{k-1}$ For different values of spectral gap $\lambda_1/\lambda_2$ and iteration number $m$, Table from (Golub and Van Loan, 2012)

eigenvalue, very often resort to the power iteration, likely due to its implementational simplicity. We showcase the power iterations relative inferiority to Lanczos with the following convergence theorems

**Theorem 1.** *Let $H^{P\times P}$ be a symmetric matrix with eigenvalues $\lambda_1 \geq .. \geq \lambda_P$ and corresponding orthonormal eigenvectors $z_1, ..z_P$. If $\theta_1 \geq .. \geq \theta_m$ are the eigenvalues of the matrix $T_m$ obtained after $m$ Lanczos steps and $q_1, ...q_m$ the corresponding Ritz eigenvectors then*

$$
\begin{aligned}
\lambda_1 \geq \theta_1 &\geq \lambda_1 - \frac{(\lambda_1 - \lambda_n)\tan^2(\theta_1)}{(c_{m-1}(1+2\rho_1))^2} \\
\lambda_P \leq \theta_k &\leq \lambda_m + \frac{(\lambda_1 - \lambda_n)\tan^2(\theta_1)}{(c_{m-1}(1+2\rho_1))^2}
\end{aligned}
\tag{6}
$$

*where $c_m$ is the Chebyshev polyomial of order $k$. $\cos\theta_1 = |q_1^T z_1|$ & $\rho_1 = (\lambda_1 - \lambda_2)/(\lambda_2 - \lambda_n)$*

*Proof.* see (Golub and Van Loan, 2012). □

**Theorem 2.** *Assuming the same notation as in Theorem 1, after $m$ power iteration steps the corresponding extremal eigenvalue estimate is lower bounded by*

$$
\lambda_1 \geq \theta_1 \geq \lambda_1 - (\lambda_1 - \lambda_n)\tan^2(\theta_1)\left(\frac{\lambda_2}{\lambda_1}\right)^{2m-1}
\tag{7}
$$

From the rapid growth of orthogonal polynomials such as Chebyshev, we expect Lanczos superiority to significantly emerge for larger spectral gap and iteration number. To verify this experimentally, we collect the non identical terms in the equations 6 and 7 of the lower bounds for $\lambda_1$ derived by Lanczos and Power iteration and denote them $L_{k-1}$ and $R_{k-1}$ respectively. For different values of $\lambda_1/\lambda_2$ and iteration number $m$ we give the ratio of these two quatities in Table 6. As can be clearly seen, the Lanczos lower bound is always closer to the true value, this improves with the iteration number $m$ and its relative edge is reduced if the spectral gap is decreased.

### B.1 THE PROBLEM OF MOMENTS: SPECTRAL DENSITY ESTIMATION USING LANCZOS

In this section we show that that the Lanczos Tri-Diagonal matrix corresponds to an orthogonal polynomial basis which matches the moments of $v^T H^m v$ and that when $v$ is a zero mean random vector with unit variance, this corresponds to the moment of the underlying spectral density.

**Stochastic trace estimation** Using the expectation of quadratic forms, for zero mean, unit variance random vectors

$$
\begin{aligned}
\mathbb{E}_v \text{Tr}(v^T H^m v) = \text{Tr}\mathbb{E}_v(vv^T H^m) &= \text{Tr}(H^m) \\
&= \sum_{i=1}^{P} \lambda_i^m = P \int_{\lambda \in \mathcal{D}} \lambda^m d\mu(\lambda)
\end{aligned}
\tag{8}
$$

where we have used the linearity of trace and expectation. Hence in expectation over the set of random vectors, the trace of the inner product of $v$ and $H^m v$ is equal to the $m$'th moment of the spectral density of $H$.

**Lanczos-Stieltjes**   The Lanczos tri-diagonal matrix $T$ can be derived from the Moment matrix $M$, corresponding to the discrete measure $d\alpha(\lambda)$ satisfying the moments $\mu_i = v^T H^i v = \int \lambda^i d\alpha(\lambda)$ (Golub and Meurant, 1994)

$$
M = \begin{bmatrix}
1 & v^T H v & \ldots & v^T H^{m-1} v \\
v^T H v & v^T H^2 v & \ddots & \vdots \\
\vdots & \ddots & \ddots & \vdots \\
v^T H^{m-1} v & \ldots & \ldots & v^T H^{2m-2} v
\end{bmatrix}
$$

and hence for a zero mean unit variance initial seed vector, the eigenvector/eigenvalue pairs of $T$ contain information about the spectral density of $H$ as shown in section B.1. This is given by the following Theorem

**Theorem 3.** *The eigenvalues of $T_k$ are the nodes $t_j$ of the Gauss quadrature rule, the weights $w_j$ are the squares of the first elements of the normalized eigenvectors of $T_k$*

*Proof.* See Golub and Meurant (1994) □

A quadrature rule is a relation of the form,

$$
\int_a^b f(\lambda) d\mu(\lambda) = \sum_{j=1}^M \rho_j f(t_j) + R[f] \tag{9}
$$

for a function $f$, such that its Riemann-Stieltjes integral and all the moments exist on the measure $d\mu(\lambda)$, on the interval $[a, b]$ and where $R[f]$ denotes the unknown remainder. The first term on the RHS of equation 9 using Theorem 3 can be seen as a discrete approximation to the spectral density matching the first $m$ moments $v^T H^m v$ (Golub and Meurant, 1994; Golub and Van Loan, 2012)

For $n_v$ starting vectors, the corresponding discrete spectral density is given as

$$
p(\lambda) = \frac{1}{n_v} \sum_{l=1}^{n_v} \left( \sum_{k=1}^m (\tau_k^{(l)})^2 \delta(\lambda - \lambda_k^{(l)}) \right), \tag{10}
$$

where $\tau_k^{(l)}$ corresponds to the first entry of the eigenvector of the $k$-th eigenvalue, $\lambda_k$, of the Lanczos tri-diagonal matrix, $T$, for the $l$-th starting vector (Ubaru and Saad; Lin et al., 2016).

## B.2   COMPUTATIONAL COMPLEXITY

For large matrices, the computational complexity of the algorithm depends on the Hessian vector product, which for neural networks is $\mathcal{O}(mNP)$ where $P$ denotes the number of parameters in the network, $m$ is the number of Lanczos iterations and $N$ is the number of data-points. The full re-orthogonalisation adds two matrix vector products, which is of cost $\mathcal{O}(m^2 P)$, where typically $m^2 \ll N$. Each random vector used can be seen as another full run of the Lanczos algorithm, so for $d$ random vectors the total complexity is $\mathcal{O}(dmP(N + m))$

**Importance of keeping orthogonality**   The update equations of the Lanczos algorithm lead to a tri-diagonal matrix $T = \mathbb{R}^{m \times m}$, whose eigenvalues represent the approximated eigenvalues of the matrix $H$ and whose eigenvectors, when projected back into the the Krylov-subspace, $\mathcal{K}(H, v)$, give the approximated eigenvectors of $H$. In finite precision, it is known (Meurant and Strakoš, 2006) that the Lanczos algorithm fails to maintain orthogonality between its Ritz vectors, with corresponding convergence failure. In order to remedy this, we re-orthonormalise at each step (Bai et al., 1996) (as shown in line 9 of Algorithm 1) and observe a high degree of orthonormality between the Ritz eigenvectors. Orthonormality is also essential for achieving accurate spectral resolution as the Ritz value weights are given by the squares of the first elements of the normalised eigenvectors. For the practitioner wishing to reduce the computational cost of maintaining orthogonality, there exist more elaborate schemes (Meurant and Strakoš, 2006; Golub and Meurant, 1994).

We also explicitly debunk some key persistent myths, given below.

---

**Algorithm 1** Lanczos Algorithm

---
1: **Input:** Hessian vector product $\{\boldsymbol{Hv}\}$, number of steps $m$
2: **Output:** Ritz eigenvalue/eigenvector pairs $\{\lambda_i, \boldsymbol{u}_i\}$ & quadrature weights $\tau_i$
3: Set $\boldsymbol{v} := \boldsymbol{v}/\sqrt{(\boldsymbol{v}^T\boldsymbol{v})}$
4: Set $\beta := 0$, $\boldsymbol{v}_{old} := \boldsymbol{v}$
5: Set $\boldsymbol{V}(:, 1) := \boldsymbol{v}$
6: **for** $j$ in $1, .., m$ **do**
7: $\quad \boldsymbol{w} = \boldsymbol{Hv} - \beta\boldsymbol{v}_{old}$
8: $\quad \boldsymbol{T}(j, j) = \alpha = \boldsymbol{w}^T\boldsymbol{v}$
9: $\quad w = \boldsymbol{w} - \alpha\boldsymbol{w} - \boldsymbol{VV}^T\boldsymbol{w}$
10: $\quad \beta = \sqrt{\boldsymbol{w}^T\boldsymbol{w}}$
11: $\quad \boldsymbol{v}_{old} = \boldsymbol{v}$
12: $\quad \boldsymbol{v} = \boldsymbol{w}/\beta$
13: $\quad \boldsymbol{V}(:, j + 1) = \boldsymbol{v}$
14: $\quad \boldsymbol{T}(j, j + 1) = \boldsymbol{T}(j + 1, 1) = \beta$
15: **end for**
16: $\{\lambda_i, \boldsymbol{e}_i\} = eig(\boldsymbol{T})$
17: $\boldsymbol{u}_i = \boldsymbol{Ve}_i$
18: $\tau_i = (\boldsymbol{e}_i^T[1, 0, 0...0])^2$

---

- We can learn the negative and interior eigenvalues by shifting and inverting the matrix sign $\boldsymbol{H} \to -\boldsymbol{H} + \mu\boldsymbol{I}$
- Lanczos learns the largest $m$ $[\lambda_i, \boldsymbol{u}_i]$ pairs of $\boldsymbol{H} \in \mathbb{R}^{P \times P}$ with high probability (Dauphin et al., 2014)

Since these two related beliefs are prevalent, we disprove them explicitly in this section, with Theorems 4 and 5.

**Theorem 4.** *The shift and invert procedure* $\boldsymbol{H} \to -\boldsymbol{H} + \mu\boldsymbol{I}$*, changes the Eigenvalues of the Tridiagonal matrix* $\boldsymbol{T}$ *(and hence the Ritz values) to* $\lambda_i = -\lambda_i + \mu$

*Proof.* Following the equations from Algorithm 1

$$
\begin{aligned}
\boldsymbol{w}_1^T &= (-\boldsymbol{H} + \mu\boldsymbol{I})\boldsymbol{v}_1 \ \& \ \alpha_1 = \boldsymbol{v}_1^T\boldsymbol{Hv}_1 + \mu\boldsymbol{I} \\
\boldsymbol{w}_2 &= \boldsymbol{w}_1 - \alpha_1\boldsymbol{v}_1 = (\boldsymbol{H} + \mu\boldsymbol{I})\boldsymbol{v}_1 - (\boldsymbol{v}_1^T\boldsymbol{Hv}_1 + \mu\boldsymbol{I})\boldsymbol{v}_1 \\
\boldsymbol{w}_2 &= (\boldsymbol{H} - \boldsymbol{v}_1^T\boldsymbol{Hv}_1)\boldsymbol{v}_1 \ \& \ \boldsymbol{v}_2 = \boldsymbol{w}_2/||\boldsymbol{w}_2|| \\
\alpha_2 &= \boldsymbol{v}_2^T(-\boldsymbol{H} + \mu\boldsymbol{I})\boldsymbol{v}_2 = -\boldsymbol{v}_2^T\boldsymbol{Hv}_2 + \mu \\
\beta_2 &= ||\boldsymbol{w}_2||
\end{aligned}
\tag{11}
$$

Assuming this for $m - 1$, and repeating the above steps for $m$ we prove by induction and finally arrive at the modified tridiagonal Lanczos matrix $\tilde{\boldsymbol{T}}$

$$
\begin{aligned}
\tilde{\boldsymbol{T}} &= -\boldsymbol{T} + \mu\boldsymbol{I} \\
\tilde{\lambda}_i &= -\lambda_i + \mu \ \forall 1 \le i \le m
\end{aligned}
\tag{12}
$$

*Remark.* No new Eigenvalues of the matrix $\boldsymbol{H}$ are learned. Although it is clear that the addition of the identity does not change the Krylov subspace, such procedures are commonplace in code pertaining to papers attempting to find the *smallest eigenvalue*. This disproves the first misconception.

**Theorem 5.** *For any matrix* $\boldsymbol{H} \in \mathbb{R}^{P \times P}$ *such that* $\lambda_1 > \lambda_2 > ..... > \lambda_P$ *and* $\sum_{i=1}^m \lambda_i < \sum_{i=m+1}^P \lambda_i$ *in expectation over the set of random vectors* $\boldsymbol{v}$ *the* $m$ *eigenvalues of the Lanczos Tridiagonal matrix* $\boldsymbol{T}$ *do not correspond to the top* $m$ *eigenvalues of* $\boldsymbol{H}$

*Proof.* Let us consider the matrix $\tilde{\boldsymbol{H}} = \boldsymbol{H} - \frac{\lambda_{m+1} + \lambda_m}{2}\boldsymbol{I}$,

$$
\begin{cases}
\lambda_i > 0, & \forall i \le m \\
\lambda_i < 0, & \forall i > m
\end{cases}
\tag{13}
$$

Under the assumptions of the theorem, $\text{Tr}(\tilde{\boldsymbol{H}}) < 0$ and hence by Theorem 3 and Equation 8 there exist no $w_i > 0$ such that

$$\sum_{i=1}^{m} w_i \lambda_i^k = \frac{1}{P} \sum_{i=1}^{P} \lambda_i^k \forall \quad 1 \le k \le m \tag{14}$$

is satisfied for $k = 1$, as the LHS is manifestly positive and the RHS is negative. By Theorem 4 this holds for the original matrix $\boldsymbol{H}$. $\qquad\square$

*Remark.* Given that Theorem 5 is satisfied over the expectation of the set of random vectors, which by the CLT is realised by Monte Carlo draws of random vectors as $d \to \infty$ the only way to really span the top $m$ eigenvectors is to have selected a vector which lies in the $m$ dimensional subspace of the $P$ dimensional problem corresponding to those vectors, which would correspond to knowing those vectors *a priori*, defeating the point of using Lanczos at all.

*Remark.* Given the relationship between the moments of the spectral density and the expectation over the set of random vectors, one may be curious to ask how we expect the deviation to be depending on the number of random vectors actually used in practice. Whilst usually packages using Lanczos use a number of random vectors (increasing the corresponding computational cost), in Appendix C we demonstrate that we expect the difference to reduce as we increase the problem dimension $P$. This informs our choice of only using a single random vector in our package, we compare results for different random vectors in Appendix C and find minimal differences.

## C    EFFECT OF VARYING RANDOM VECTORS

Given that the proofs for the moments of Lanczos matching those of the underlying spectral density, are true over the expectation over the set of random vectors and in practice we only use a Monte Carlo average of random vectors, or in our experiments using stem plots, just a single random vector. We justify this with the following Lemma

**Lemma 1.** *Let* $\boldsymbol{u} \in \mathbb{R}^{P \times 1}$ *random vector, where* $\boldsymbol{u}_i$ *is zero mean and unit variance and finite 4'th moment* $\mathbb{E}[\boldsymbol{u}_i^4] = m_4$. *Then for* $\boldsymbol{H} \in \mathbb{R}^{P \times P}$, *then*

$$i) \mathbb{E}[\boldsymbol{u}^T \boldsymbol{H} \boldsymbol{u}] = \text{Tr}\,\boldsymbol{H}$$
$$ii) \text{Var}[\boldsymbol{u}^T \boldsymbol{H} \boldsymbol{u}] \le (2 + m_4) \text{Tr}(\boldsymbol{H}^T \boldsymbol{H})$$

*Proof.*

$$\mathbb{E}[\boldsymbol{u}^T \boldsymbol{H} \boldsymbol{u}] = \sum_{i,j=1}^{P} \boldsymbol{H}_{i,j} \mathbb{E}[\boldsymbol{u}_i \boldsymbol{v}_j] = \sum_{i=1}^{P} \boldsymbol{H}_{i,i} = \text{Tr}\,\boldsymbol{H} \tag{15}$$

$$\mathbb{E}[||\boldsymbol{u}^T \boldsymbol{H} \boldsymbol{u}||^2] = \sum_{i,j} \sum_{k,l} \boldsymbol{H}_{i,j} \boldsymbol{H}_{k,l}^T \mathbb{E}[\boldsymbol{u}_i \boldsymbol{u}_j^T \boldsymbol{u}_k \boldsymbol{u}_l^T]$$

$$\sum_{i,j} \sum_{k,l} \boldsymbol{H}_{i,j} \boldsymbol{H}_{k,l}^T [\delta_{i,j}\delta_{k,l} + \delta_{i,l}\delta_{j,k} + \delta_{i,k}\delta_{j,l} + m_4 \delta_{i,j,k,l}] \tag{16}$$

$$= (\text{Tr}\,\boldsymbol{H})^2 + (2 + m_4)\text{Tr}(\boldsymbol{H}^2)$$

$\qquad\square$

*Remark.* Let us consider the signal to noise ratio for some positive definite $\boldsymbol{H} \succ c\boldsymbol{I}$

$$\left( \frac{\sqrt{\text{Var}[\boldsymbol{u}^T \boldsymbol{H} \boldsymbol{u}]}}{\mathbb{E}[\boldsymbol{u}^T \boldsymbol{H} \boldsymbol{u}]} \right)^2 \propto \frac{1}{1 + \frac{\sum_{i \ne j}^{P} \lambda_i \lambda_j}{\sum_k^P \lambda_k^2}} = \frac{1}{1 + \frac{P-1 \langle \lambda_i \lambda_j \rangle}{\langle \lambda_k^2 \rangle}}$$

$$\le \frac{1}{1 + \frac{P-1}{\kappa^2}} \tag{17}$$

where $\langle .. \rangle$ denotes the arithmetic average. For the extreme case of all eigenvalues being identical, the condition number $\kappa = 1$ and hence this reduces to $1/P \to 0$ in the $P \to \infty$ limit, whereas for a rank-1 matrix, this ratio remains 1. For the MP density, which well models neural network spectra, $\kappa$ is not a function of $P$ as $P \to \infty$ and hence we also expect this benign dimensional scaling to apply.

We verify this high dimensional result experimentally, by running spectral visualisation but using two different random vectors. We plot the results in Figure 5. We find both figures 5a & 5b to be close to visually indistinguishable. There are minimal differences in the extremal eigenvalues, with former giving $\{\lambda_1, \lambda_n\} = \{6.8885, -0.0455\}$ and the latter $\{6.8891, -0.0456\}$, but the degeneracy at 0, bulk, triplet of outliers at 2.27 and the large outlier at 6.89 is unchanged.

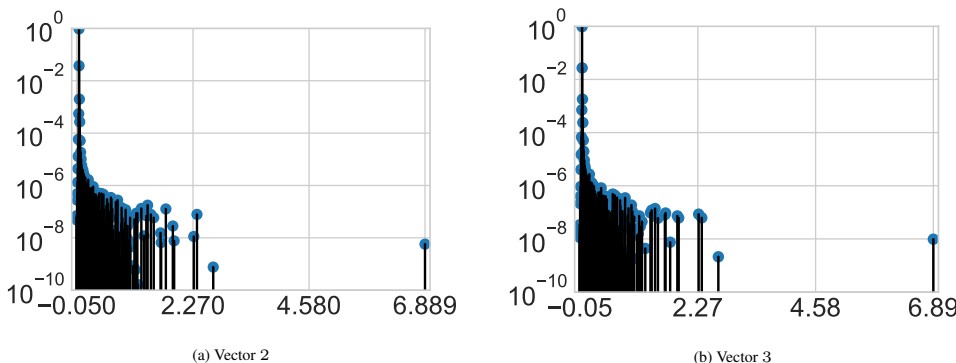

(a) Vector 2

(b) Vector 3

Figure 5: VGG16 Epoch 300 end of training Lanczos stem plot for different random vectors

### C.1 WHY WE DON'T KERNEL SMOOTH

Concurrent work, which has also used Lanczos with the Pearlmutter trick to learn the Hessian (Yao et al., 2018; Ghorbani et al., 2019), typically uses $n_v$ random vectors and then uses kernel smoothing, to give a final density. In this section we argue that beyond costing a factor of $n_v$ more computationally, that the extra compute extended in order to get more accurate moment estimates, which we already argued in Section C are asymptotically error free, is wasted due to the kernel smoothing (Granziol et al., 2019). The smoothed spectral density takes the form:

$$\tilde{p}(\lambda) = \int k_\sigma(\lambda - \lambda')p(\lambda')d\lambda' = \sum_{i=1}^n w_i k_\sigma(\lambda - \lambda_i) \tag{18}$$

We make some assumptions regarding the nature of the kernel function, $k_\sigma(\lambda - \lambda_i)$, in order to prove our main theoretical result about the effect of kernel smoothing on the moments of the underlying spectral density. Both of our assumptions are met by (the commonly employed) Gaussian kernel.

**Assumption 1.** *The kernel function $k_\sigma(\lambda - \lambda_i)$ is supported on the real line $[-\infty, \infty]$.*

**Assumption 2.** *The kernel function $k_\sigma(\lambda - \lambda_i)$ is symmetric and permits all moments.*

**Theorem 6.** *The $m$-th moment of a Dirac mixture $\sum_{i=1}^n w_i\delta(\lambda - \lambda_i)$, which is smoothed by a kernel $k_\sigma$ satisfying assumptions 1 and & 2, is perturbed from its unsmoothed counterpart by an amount $\sum_{i=1}^n w_i \sum_{j=1}^{r/2} \binom{r}{2j} \mathbb{E}_{k_\sigma(\lambda)}(\lambda^{2j})\lambda_i^{m-2j}$, where $r = m$ if $m$ is even and $m - 1$ otherwise. $\mathbb{E}_{k_\sigma(\lambda)}(\lambda^{2j})$ denotes the $2j$-th central moment of the kernel function $k_\sigma(\lambda)$.*

*Proof.* The moments of the Dirac mixture are given as,

$$\langle \lambda^m \rangle = \sum_{i=1}^n w_i \int \delta(\lambda - \lambda_i)\lambda^m d\lambda = \sum_{i=1}^n w_i \lambda_i^m. \tag{19}$$

The moments of the modified smooth function (Equation equation 18) are

$$\langle \tilde{\lambda}^m \rangle = \sum_{i=1}^n w_i \int k_\sigma(\lambda - \lambda_i)\lambda^m d\lambda$$

$$= \sum_{i=1}^n w_i \int k_\sigma(\lambda')(\lambda' + \lambda_i)^m d\lambda' \tag{20}$$

$$= \langle \lambda^m \rangle + \sum_{i=1}^n w_i \sum_{j=1}^{r/2} \binom{r}{2j} \mathbb{E}_{k_\sigma(\lambda)}(\lambda^{2j})\lambda_i^{m-2j}.$$

We have used the binomial expansion and the fact that the infinite domain is invariant under shift reparametarization and the odd moments of a symmetric distribution are 0. □

*Remark.* The above proves that kernel smoothing alters moment information, and that this process becomes more pronounced for higher moments. Furthermore, given that $w_i > 0$, $\mathbb{E}_{k_\sigma(\lambda)}(\lambda^{2j}) > 0$ and (for the GGN $lambda_i > 0$, the corrective term is manifestly positive, so the smoothed moment estimates are biased.

## D   LOCAL LOSS LANDSCAPE

The Lanczos algorithm with enforced orthogonality initialised with a random vector gives a moment matched discrete approximation to the Hessian spectrum. However this information is local to the point in weight space $w$ and the quadratic approximation may break down within the near vicinity. To investigate this, we use the **loss landscape visualisation** function of our package: We display this for the VGG-16 on CIFAR-100 in Figure 8. We see for the training loss 6a that the eigenvector corresponding to the largest eigenvalue $\lambda = 6.88$ only very locally corresponds to the sharpest increase in loss for the training, with other extremal eigenvectors, corresponding to the eigenvalues $\lambda = \{2.67, 2.35\}$ overtaking it in loss change relatively rapidly. Interestingly for the testing loss, all the extremal eigenvectors change the loss much more rapidly, contradicting previous assertions that the test loss is a "shifted" version of the training loss (He et al., 2019; Izmailov et al., 2018). We do however note some small asymettry between the changes in loss along the opposite ends of the eigenvectors. The flat directions remain flat locally and some of the eigen-vectors corresponding to negative values correspond to decreases in test loss. We include the code in **??**

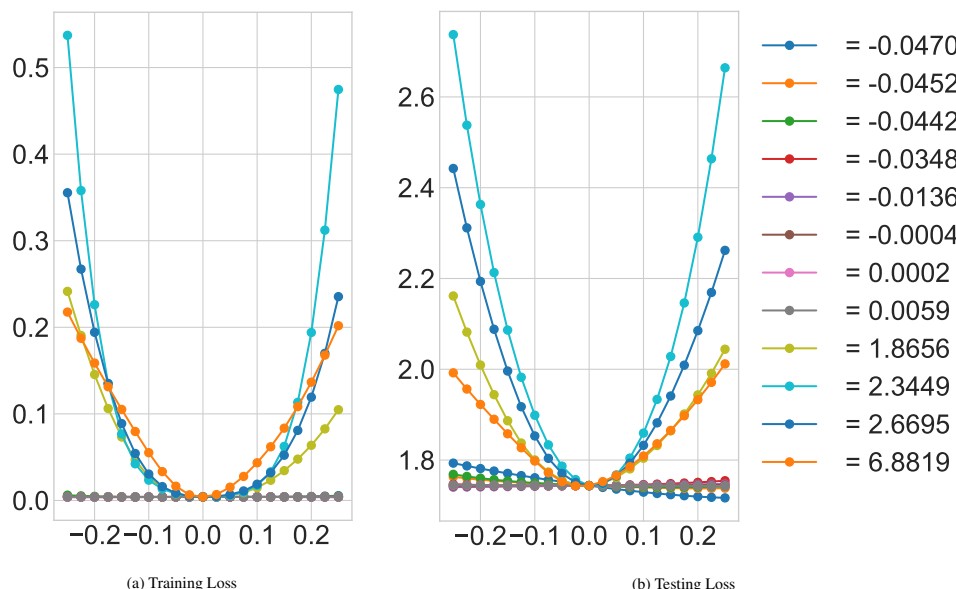

(a) Training Loss          (b) Testing Loss

Figure 6: VGG-16 CIFAR-100 Loss surface visualised along 6 negative and position eigenvalues

CIFAR-10 DATASET

To showcase the ability of our software to handle multiple datasets we display the Hessian of the VGG-16 trained in an identical fashion as its CIFAR-100 counterpart of CIFAR-10 in Figure **??**, along with the a plot of a selection of Ritz vectors traversing the training loss surface in Figure 8a and testing loss surface in Figure 8b along with also the training accuracy surface (Figure 9a) and testing accuracy surface (Figure 9b).

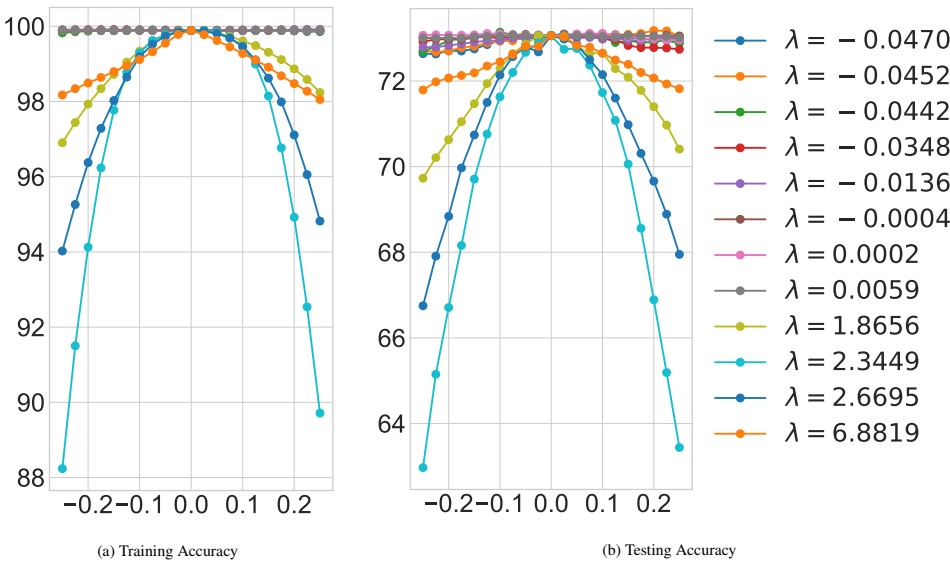

(a) Training Accuracy

(b) Testing Accuracy

Figure 7: VGG-16 CIFAR-100 Accuracy surface visualised along 6 negative and position eigenvalues

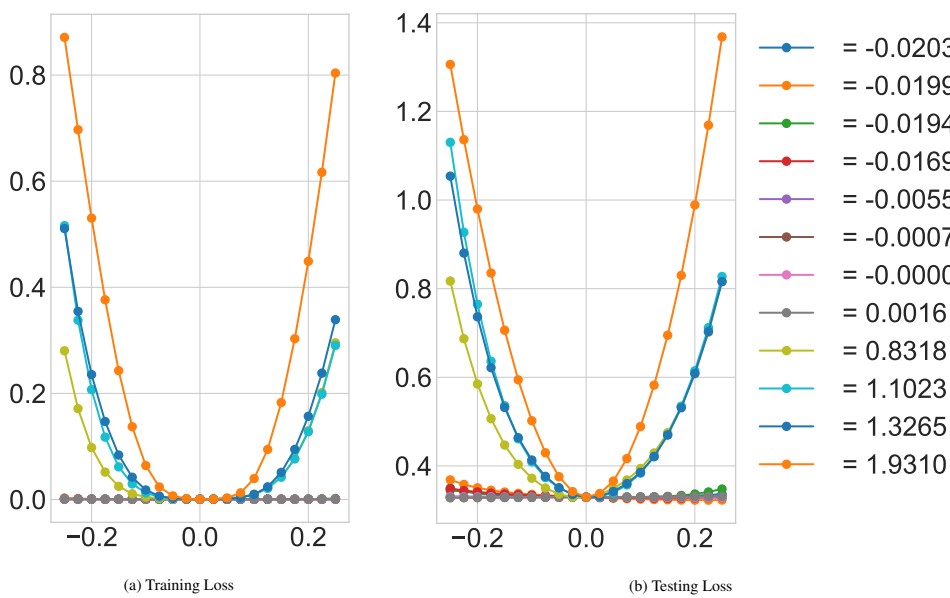

(a) Training Loss

(b) Testing Loss

Figure 8: VGG-16 CIFAR-10 Loss surface visualised along 6 negative and position eigenvalues

# E    EXAMPLES ON SMALL RANDOM MATRICES

In this section, we use some examples on small random matrices to showcase the power of our package that uses the Lanczos algorithm with random vectors to learn the spectral density. Here, we look at known random matrices with elements drawn from specific distributions which converge to known spectral densities in the asymptotic limit. Here we consider **Wigner Ensemble** (Wigner, 1993) and the **Marcenko Pastur** (Marchenko and Pastur, 1967), both of which are extensively used in simulations or theoretical analyses of deep neural network spectra (Pennington and Bahri, 2017; Choromanska et al., 2015a; Anonymous, 2020).

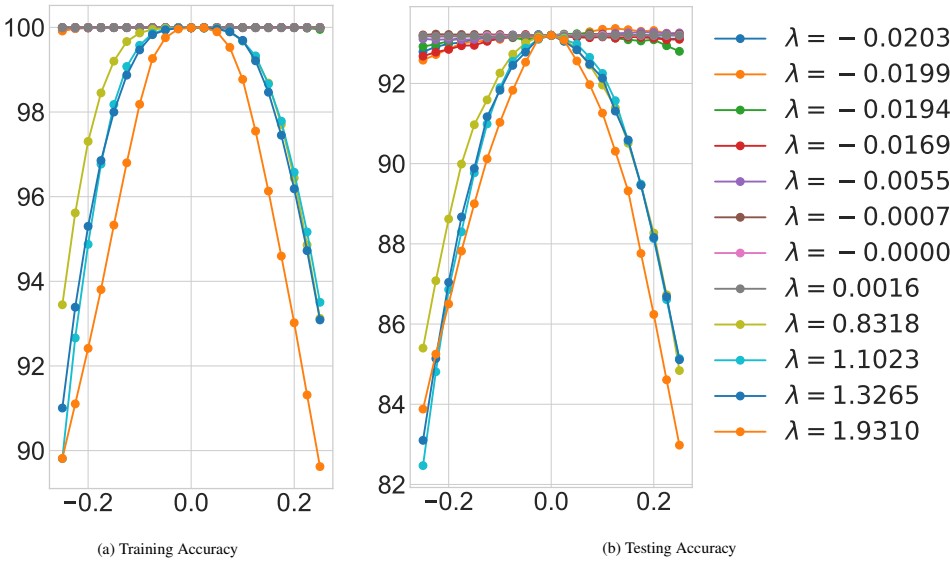

(a) Training Accuracy    (b) Testing Accuracy

Figure 9: VGG-16 CIFAR-10 Accuracy surface visualised along negative and positive eigenvalues

### E.1 WIGNER MATRICES

Wigner matrices can be defined in Definition E.1, and their distributions of eigenvalues are governed by the semi-circle distribution law (Theorem 7).

**Definition E.1.** *Let $\{Y_i\}$ and $\{Z_{ij}\}_{1 \leq i \leq j}$ be two real-valued families of zero mean, i.i.d random variables, Furthermore suppose that $\mathbb{E}Z_{12}^2 = 1$ and for each $k \in \mathbb{N}$*

$$max(E|Z_{12}^k, E|Y_1|^k) < \infty \tag{21}$$

*Consider a $P \times P$ symmetric matrix $M_P$, whose entries are given by*

$$\begin{cases} M_P(i,i) = Y_i \\ M_P(i,j) = Z_{ij} = M_P(j,i), & \text{if } x \geq 1 \end{cases} \tag{22}$$

*The Matrix $M_P$ is known as a real symmetric Wigner matrix.*

**Theorem 7.** *Let $\{M_P\}_{P=1}^{\infty}$ be a sequence of Wigner matrices, and for each $P$ denote $X_P = M_P/\sqrt{P}$. Then $\mu_{X_P}$, converges weakly, almost surely to the semi circle distribution,*

$$\sigma(x)dx = \frac{1}{2\pi}\sqrt{4 - x^2}\mathbf{1}_{|x| \leq 2} \tag{23}$$

For our experiments, we generate random matrices $\boldsymbol{H} \in \mathbb{R}^{P \times P}$ with elements drawn from the distribution $\mathcal{N}(0,1)$ for $P = \{225, 10000\}$ and plot histogram of the spectra found by eigendecomposition, along with the predicted Wigner density (scaled by a factor of $\sqrt{P}$) in Figures 10b & 10d and compare them along with the discrete spectral density approximation learned by lanczos in $m = 30$ steps using a single random vector $d = 1$ in Figures 10a & 10c. It can be seen that even for a small number of steps $m \ll P$ and a single random vector, Lanczos impressively captures not only the support of the eigenvalue spectral density but also its shape. We note as discussed in section B.2 that the 30 Ritz values here do not span the top 30 eigenvalues even approximately.

### E.2 MARCENKO-PASTUR

An equally important limiting law for the limiting spectral density of many classes of matrices constrained to be positive definite, such as covariance matrices, is the Marcenko-Pastur law (Marchenko and Pastur, 1967). Formally, given a matrix $\boldsymbol{X} \in \mathbb{R}^{P \times T}$ with i.i.d zero mean entires with variance $\sigma^2 < \infty$. Let $\lambda_1 \geq \lambda_2, ... \geq \lambda_P$ be eigenvalues of $\boldsymbol{Y}_n = \frac{1}{T}\boldsymbol{X}\boldsymbol{X}^T$. The random measure $\mu_P(A) = \frac{1}{P}\#\{\lambda_j \in A\}, A \in \mathbb{R}$

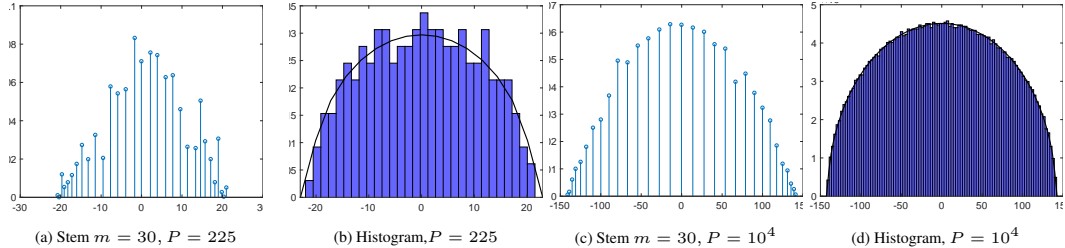

Figure 10: Lanczos stem plot for a single random vector with $m = 30$ steps compared to actual eigenvalue histogram for matrices of the form $\boldsymbol{H} \in \mathbb{R}^{P \times P}$, where each element is a drawn from a normal distribution with unit variance, converging to the Wigner semi circle.

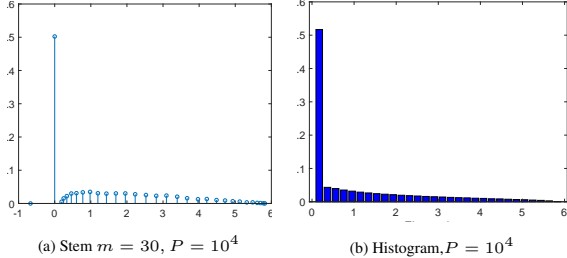

Figure 11: Lanczos stem plot for a single random vector with $m = 30$ steps compared to actual eigenvalue histogram for matrices of the form $\boldsymbol{H} \in \mathbb{R}^{P \times P}$, where $\boldsymbol{H} = \boldsymbol{X}\boldsymbol{X}^T/k$, where each element of $\boldsymbol{X}^{P \times k}$, $k = 0.5P$ is a drawn from a normal distribution with unit variance, converging to the Marcenko-Pastur distribution with $q = 0.5$.

**Theorem 8.** *Assume that $P, N \to \infty$ and the ratio $P/N \to q \in (0, \infty)$ (this is known as the Kolmogorov limit) then $\mu_P \to \mu$ in distribution where*

$$
\begin{cases}
(1 - \frac{1}{q})\mathbb{1}_{0 \in A} + \nu_{1/q}(A), & \text{if } q > 1 \\
\nu_q(A), & \text{if } 0 \leq q \leq 1
\end{cases}
\tag{24}
$$

$$
d\nu_q = \frac{\sqrt{(\lambda_+ - x)(x - \lambda_-)}}{\lambda x 2\pi\sigma^2}, \lambda_{\pm} = \sigma^2(1 \pm \sqrt{q})^2
\tag{25}
$$

Here, we construct a random matrix $\boldsymbol{X} \in \mathbb{P} \times \mathbb{T}$ with independently drawn elements from the distribution $\mathcal{N}(0, 1)$ and then form the matrix $\frac{1}{T}\boldsymbol{X}\boldsymbol{X}^T$, which is known to converge to the Marcenko-Pastur distribution. We use $P = \{225, 10000\}$ and $T = 2P$ and plot the associated histograms from full eigendecomposition in Figures 12b & 12d along with their $m = 30, d = 1$ Lanczos stem counterparts in Figures 12a & 12c. Similarly we see a faithful capturing not just of the support, but also of the general shape. We note that both for Figure 10 and Figure 12, the smoothness of the discrete spectral density for a single random vector increases significantly, even relative to the histogram.

We also run the same experiment for $P = 10000$ but this time with $T = 0.5P$ so that exactly half of the eigenvalues will be $0$. We compare the Histogram of the eigenvalues in Figure 11b against its $m = 30, d = 1$ Lanczos stem plot in Figure 11a and find both the density at the origin, along with the bulk and support to be faithfully captured.

### E.3    COMPARISON TO DIAGONAL APPROXIMATIONS

As a proxy for deep neural network spectra, often the diagonal of the matrix (Bishop, 2006) or the diagonal of a surrogate matrix, such as the Fisher information, or that implied by the values of the Adam Optimizer (Chaudhari et al., 2016) is used. We plot the true eigenvalue estimates for random matrices pertaining to both the Marcenko-Pastur (Fig. 14a) and the Wigner density (Fig. 14b) in blue, along with the Lanczos estimate in red and the diagonal approximation in yellow. We see here that the diagonal approximation in both cases, fails to adequately the support or accurately model the spectral density, whereas the lanczos estimate is nearly indistinguishable from the true binned eigenspectrum. This is of-course obvious from the mathematics of the un-normalised Wigner matrix. The diagonal elements are simply draws from the normal distribution $\mathcal{N}(0, 1)$ and so we expect the

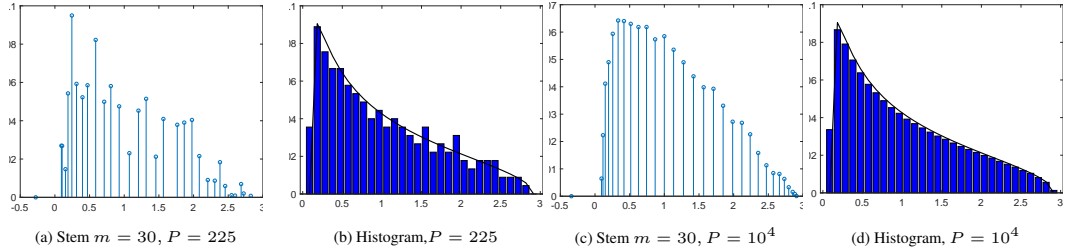

Figure 12: Lanczos stem plot for a single random vector with $m = 30$ steps compared to actual eigenvalue histogram for matrices of the form $\boldsymbol{H} \in \mathbb{R}^{P \times P}$, where $\boldsymbol{H} = \boldsymbol{X}\boldsymbol{X}^T/k$, where each element of $\boldsymbol{X}^{P \times k}$, $k = 2P$ is a drawn from a normal distribution with unit variance, converging to the Marcenko-Pastur distribution with $q = 2$.

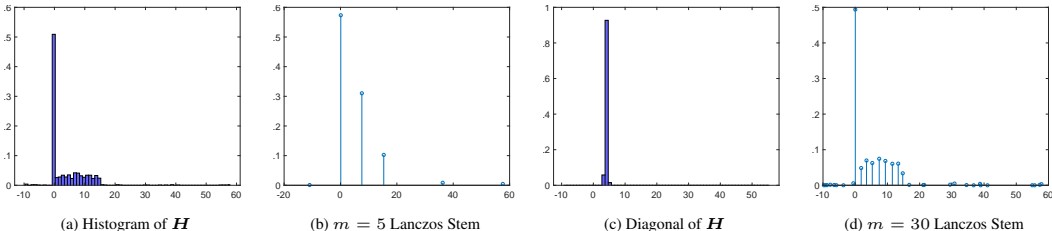

Figure 13: Generated matrices $\boldsymbol{H} \in \mathbb{R}^{1000 \times 1000}$ with known eigenspectrum and Lanczos stem plots for different values of $m = \{5, 15, 30\}$

diagonal histogram plot to approximately follow this distribution (with variance 1). However the second moment of the Wigner Matrix can be given by the Frobenius norm identity

$$\mathbb{E}\left(\frac{1}{P}\sum_i^P \lambda_i^2\right) = \mathbb{E}\left(\frac{1}{P}\sum_{i,j=1}^P \boldsymbol{H}_{i.j}^2\right) = \mathbb{E}\left(\frac{1}{P}\chi_{P^2}^2\right) = P \tag{26}$$

Similarly for the Marcenko-Pastur distribution, We can easily see that each element of $\boldsymbol{H}$ follows a chi-square distribution of $1/T\chi_T^2$, with mean 1 and variance $2/T$.

### E.4 SYNTHETIC EXAMPLE

The curvature eigenspectrum of neural network often features a large spike at zero, a right-skewed bulk and some outliers (Sagun et al., 2016; 2017).[3] In order to simulate the spectrum of a neural network, we generate a Matrix $\boldsymbol{H} \in \mathbb{R}^{1000 \times 1000}$ with 470 eigenvalues drawn from the uniform distribution from $[0, 15]$, 20 drawn from the uniform $[0, 60]$ and 10 drawn from the uniform $[-10, 0]$. The matrix is rotated through a rotation matrix $U$, i.e $\boldsymbol{H} = \boldsymbol{U}\boldsymbol{D}\boldsymbol{U}^T$ where $\boldsymbol{D}$ is the diagonal matrix consisting of the eigenvalues and the columns are gaussian random vectors which are orthogonalised using Gram-Schmidt orthogonalisation. The resulting eigenspectrum is given in a histogram in

---

[3]Some examples of this can be found in later sections on real-life neural network experiments - see Figures 15 and 5.

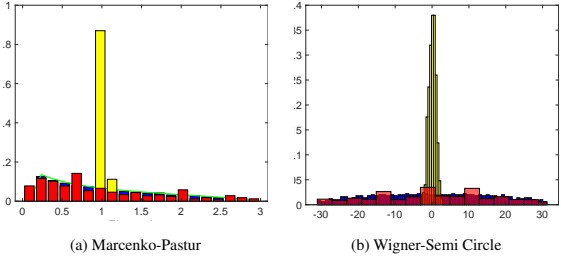

Figure 14: Two randomly generated matrices $\boldsymbol{H} \in \mathbb{R}^{500 \times 500}$ with the histogram of the true eigenvalues in blue, the Lanczos estimate $m = 30$, $d = 1$ in red and the diagonal approximation in yellow

Figure 13a and then using the same random vector, successive Lanczos stem plots for different number of iterations $m = [5, 30]$ are shown in Figure 13. Figure 13b, for a low number of steps, the degeneracy at $\lambda = 0$ is learned, as are the largest and smallest eigenvalues, some information is retained about the bulk density, but some of the outlier eigenvalues around $\lambda \approx 20$ and $\lambda \approx 30$ are completely missed out, along with all the negative outliers except the largest. For $m = 30$ even the shape of the bulk is accurately represented, as shown in Figure 13d. Here, we would like to emphasise that learning the outliers is important in the neural network context, as they relate to important properties of the network and the optimisation process (Ghorbani et al., 2019).

On the other hand, we note that the diagonal estimate in Figure 13c gives absolutely no spectral information, with no outliers shown (maximal and minimal diagonal elements being $5.3$ and $3.3$ respectively and it also gets the spectral mass at $0$ wrong. This builds on section E.3, as furthering the case against making diagonal approximations in general. In neural networks, the diagonal approximation is similar to positing no correlations between the weights. This is a very harsh assumption and usually a more reasonable assumption is to posit that the correlations between weights in the same layer are larger than between different layers, leading to a block diagonal approximation (Martens, 2016), however often when the layers have millions of parameters, full diagonal approximations are still used. (Bishop, 2006; Chaudhari et al., 2016).

### E.5 COMPARISON TO BACKPACK

For 16-layer VGG network, on the CIFAR-100 dataset the GGN diagonal computations Dangel et al. (2019) require over 125GB of GPU memory. Hence we use their Monte Carlo approximation to the GGN diagonal against both our GGN-Hessian-Lanczos spectral visualizations. We plot a histogram of the Monte Carlo approximation of the diagonal GGN (Diag-GGN) against both the Lanczos GGN (Lanc-GGN) and Lanczos Hessian (Lanc-Hess) in Figure 15. Note that as the Lanc-GGN and Lanc-Hess are displayed as stem plots (with the discrete spectral density summing to 1 as opposed to the histogram area summing to 1). We note that the Gauss-Newton approximation quite closely resembles its Hessian counterpart, capturing the majority of the bulk and the outlier eigenvectors at $\lambda_1 \approx 6.88$ and the triad near $\lambda_i \approx 2.29$. The Hessian does still have significant spectral mass on the negative axis, around $37\%$. However most of this is captured by a Ritz value at $-0.0003$, with this removed, the negative spectral mass is only $0.05\%$. However the Diag-GGN gives a very poor spectral approximation. It vastly overestimates the bulk region, which extends well beyond $\lambda \approx 1$ implied by Lanczos and adds many spurious outliers between $3$ and the misses the largest outlier of $6.88$. *Computational Cost* Using a single NVIDIA GeForce GTX 1080 Ti GPU, the Gauss-Newton takes an average $26.5$ seconds for each Lanczos iteration with the memory useage 2850Mb. Using the Hessian takes an average of $27.9$ seconds for each Lanczos iteration with 2450Mb memory usage.

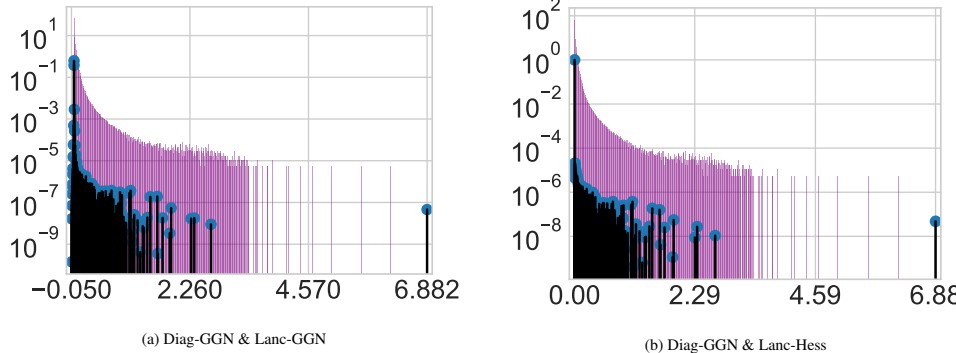

(a) Diag-GGN & Lanc-GGN

(b) Diag-GGN & Lanc-Hess

Figure 15: Diagonal Generalised Gauss-Newton monte carlo approximation (Diag-GGN) against $m = 100$ Lanczos using Gauss-Newton vector products (Lanc-GGN) or Hessian vector products (Lanc-Hess)

