# OpenReview forum: "Deep Curvature Suite"
_ICLR.cc/2021/Conference — Reject_

### Official Review · AnonReviewer2 · 2020-10-28
**Unclear contribution; supplementary material not anonymized**

**Rating:** 3
**Confidence:** 4

**Review:**

I think this paper needs to be desk-rejected since the supplementary material reveals the authors' identities in multiple places, e.g. the Readme file and the example notebook. Various .py files have an author attribute set.

This issue aside, both the package and the paper appear rushed to me and **not ready for publication**.

Package:
* Most importantly, I could not find any tests whatsoever. Of course tests do not guarantee anything, especially if implemented poorly, but I hope that nobody would trust a completely untested third-party codebase for their own research. A thorough(!) test suite would be a strict requirement from my view to even consider accepting a code library paper at a top-tier conference.
* There is no setup.py file to install the package and dependencies. I tried importing some of the modules from the root directory but got errors because of missing dependencies (e.g. backpack is not listed in the requirements.txt).
* The package is split into multiples modules with generic names such as 'core'. I would strongly suggest to put them in a joint namespace.
* What's the reasoning for including implementations of standard architectures that are available e.g. through torchvision and high-level training functions? Are those required? In general, I think that for a utility library it is best to assume that users have an established workflow of constructing and training their networks (unless the point of the library is to simplify that workflow, but this does not seem to be the case here).

Paper:
* I don't think that Section 4 fits into the flow of the paper. I would rather see that space used to go more into depth with the examples. At the moment you assume that the reader already knows why they would want to e.g. estimate the spectral density, but I think it would be worth both motivating the experiments and discussing the results.
* I think basing the code examples on actual network and data loader objects would make them more accessible. Presumably passing string arguments for model and dataset into your compute_eigenspectrum and train_network functions is limited to the architectures and datasets that you implement? How do users provide their own datasets or networks? Or is this not supported? Section 2 seems to claim otherwise.
* The font sizes and families around the code listings are extremely inconsistent, especially pages 5 and 6 are visually fairly unappealing.
* There is some discussion around the relationship to Backpack, but what about PyHessian? That package seems more closely related since it also focuses on computing spectral densities.
* Comparing first and second order optimisers on a per-epoch basis as in Figure 1 does not seem practically relevant to me, a wall clock time comparison would be of much higher interest to most potential users.
* I would recommend carefully checking the paper for spelling and language. I'm not a native speaker, but to me it seems like there are quite a few inappropriate capitalisations ("Training", "Testing"), inconsistent capitalisations ("eigenvalues" vs "Eigenvalues") and hyphenations ("eigen-decomposition").
* The reference pages are missing (after page 8).

---

### Official Review · AnonReviewer1 · 2020-10-28
**Deep Curvature Suite could potentially stimulate more research**

**Rating:** 7
**Confidence:** 4

**Review:**

###
Summary:
This paper proposes a software package to ease and provide a standard way for Hessian-related computation, both for loss analysis and second order optimization. It also provides analysis on why Lanczos algorithm is a better choice to estimate Hessian eigenvalue compared to Power Iterations. Finally, it empirically shows the advantage of using Hessian approximation that goes beyond diagonal approximation for spectrum computation.

###
Reasons for score:
I lean toward acceptance. Implementing Hessian-related computation is complex and often a bottleneck for research. Easing the implementation of those computations lower the entry barrier for second order optimization/loss analysis research and has the potential to stimulate more works in those areas.

###
 Pros:
- Address an important implementation problem
- Scale to medium size network used on CIFAR-10.

###
Cons:
- Comparisons with related work could be expanded. How does this library related to the other software that use Lanczos algorithm for Hessian computation?
- How extensive is the library? How easy is to use it with an arbitrary Pytorch model?
- It is not clear if the library would scale to neural network usually used for larger scale problem such as ImageNet.

###
Questions:
- It is not clear to me how to compute the Hessian of a Neural Network with Batch Norm using minibatch statistics due to the dependency on the other samples of the batch. Could you elaborate on this point?

---

### Official Review · AnonReviewer5 · 2020-11-06
**Useful library, paper needs major revision**

**Rating:** 4
**Confidence:** 4

**Review:**

### Summary
The paper presents a library for second-order analysis of the optimization of models implemented with PyTorch and potentially having millions of parameters. The library offers tools for easily computing and visualizing Hessian, curvature, loss landscape, and runnning simple statistics, for instance for studying the properties of local minima.
Compared to existing tools, the proposed library is more complete and accurate, , as demonstrated by example analyses, and scales well with the number of model parameters and the dataset size.
The library itself seems very useful, however the paper needs major revision in order to be publishable.

### Presentation
The paper is generally poorly written. Many concepts are not defined or not well enough, for instance:
1. input independence, page 2, by which it is not clear if independence between features or between samples is meant;
2. number of moments, page 3, not defined;
3. curvature, by which it should be specified earlier it is meant the loss curvature.

To improve clarity, the paper should provide a comprehensive list of tools implemented in the library and give examples of why one would need to use each of them. The library's tools are described throughout the paper, but they are too scattered to have a complete view of the library.
Moreover, the Lanczos algorithm, which is the central for the library, should be reported.

Finally, the paper misses important parts, such as the reference page, a paragraph in page 5 and the cited and not reported Algorithm 1; it also has many typos; notations and domains are not provided for most equations and it is not clear why Equation (3) does not depend on the loss L.

The current presentation doesn't meet the standards for a conference paper.

### UPDATE
The presentation of the paper is now convincing, with the background, contributions and concepts clearly stated. I hence increased my score.

However, I agree with reviewer 2 that the library is not properly tested (I understand it can be hard to test the whole computation, but unit tests could be easily provided) and that it would have a higher impact if it were more modular, so that a user could easily add the loss analysis directly in her workflow.
Moreover, I also think that the paper should be rejected given that the first submission wasn't anonymized and the paper wasn't in a state of being submitted.

---

### Official Review · AnonReviewer4 · 2020-11-06
**Well-written paper and can be a good contribution to the community.**

**Rating:** 6
**Confidence:** 3

**Review:**

Summary:
- This paper introduces a package for computing the second-order information of neural networks based on the Lanczos algorithm. The authors showcase the usages of the package with 1) visualizing the eigenspectrum of the curvature matrix; 2)visualizing the loss surface along with a specific direction, and 3) comparisons between different optimizers. Also, the authors claimed that they address several misconceptions about the Lanczos algorithm.

Overall:

This paper is well-written and easy to follow. I believe that the developed package can be a good contribution to the machine learning community, especially for people working on second-order optimization and understanding the training dynamics and generalization performance of deep neural networks. However, I still have the following questions:

-  Lanczos algorithm suffers from numerical instability. How do you deal with this issue? Can you provide more details about this?

- I would suggest the authors move Section 4 to the appendix, considering the main scope of the paper is introducing a new machine learning package. In the meantime, the authors should highlight more on the difference between your package and the existing implementations. For example, you can add a table to summarize the features of all the related packages/implementations. Also, a table for comparing the running memory cost and running time cost will be very helpful.

Minors:
-  as well as the commonly used `Generalsed` Gauss Newton -> as well as the commonly used `Generalised` Gauss-Newton

- The main interface functions are `organsed` as followed: -> The main interface functions are `organized` as following:

- Krylov subspace K (H, v) = span{v,`H^v`,H2v...} is orthogonalised -> Krylov subspace K (H, v) = span{v,`Hv`,H2v...} is orthogonalised

- the reference is missing in the main pdf.


Rating:
- This paper did a good job of introducing the package with detailed examples. Also, this package will definitely ease the effort of researchers in related areas to compute the second-order information of the neural networks. In the meantime, I still have the concerns mentioned above. So, my rating is weak acceptance at the current stage.

---

### Decision · Program_Chairs · 2021-01-07
**Final Decision**

**Decision:**

Reject

**Comment:**

This paper proposes a software package to ease and provide a standard way for Hessian-related computation, both for loss analysis and second order optimization. I think all reviewers agree with the usefulness of this work but differ in their assessment whether this work is ready for publication. Given the emphasize on providing a software package I share the view that careful testing and support of usability is important. While there is quite some spread in the scores I think in this case the average score is an appropriate way to compensate for subjective differences between reviewers. I think it is justified to encourage the authors to invest a bit more work to turn this into a fully convincing contribution.